# Fast Similarity Search via Optimal Sparse Lifting

**Wenye Li**[1,2,*]**, Jingwei Mao**[1]**, Yin Zhang**[1]**, Shuguang Cui**[1,2]
[1] The Chinese University of Hong Kong, Shenzhen, China
[2] Shenzhen Research Institute of Big Data, Shenzhen, China
`{wyli,yinzhang,shuguangcui}@cuhk.edu.cn, 216019005@link.cuhk.edu.cn`

## Abstract

Similarity search is a fundamental problem in computing science with various applications and has attracted significant research attention, especially in large-scale search with high dimensions. Motivated by the evidence in biological science, our work develops a novel approach for similarity search. Fundamentally different from existing methods that typically reduce the dimension of the data to lessen the computational complexity and speed up the search, our approach projects the data into an even higher-dimensional space while ensuring the sparsity of the data in the output space, with the objective of further improving precision and speed. Specifically, our approach has two key steps. Firstly, it computes the *optimal sparse lifting* for given input samples and increases the dimension of the data while approximately preserving their pairwise similarity. Secondly, it seeks the *optimal lifting operator* that best maps input samples to the *optimal sparse lifting*. Computationally, both steps are modeled as optimization problems that can be efficiently and effectively solved by the *Frank-Wolfe* algorithm. Simple as it is, our approach has reported significantly improved results in empirical evaluations, and exhibited its high potentials in solving practical problems.

## 1  Introduction

Similarity search refers to the problem of finding a subset of objects which are similar to a given query from a specific dataset. As a fundamental problem in computing science, it has various applications in information retrieval, pattern classification, data clustering, etc., and has attracted significant research attention in the literature [21, 9].

More specifically and of particular research interest, recent work in similarity search focuses on the large-scale high-dimensional problems. To lessen the computational complexity, a popular approach is to firstly reduce the dimension of the data, and then apply the nearest neighbor search or the space partitioning methods effectively on the reduced data. To efficiently reduce the dimension of large volume datasets, the *locality sensitive hashing* method is widely used [11, 3, 7], with quite successful results.

Very recently, with biological evidence from the fruit fly's olfactory circuit, people have shown the possibility of increasing the data dimension instead of reducing it as a general hashing scheme. For example, the *fly* algorithm projects each input data sample to a higher-dimensional output space with a random sparse binary matrix. Then after competitive learning, the algorithm returns a sparse binary vector in the output space. Comparing with the *locality sensitive hashing* method, similarity search based on the sparse binary vectors has reported improved precision and speed in a series of empirical studies [8].

Motivated by the biological evidence and the idea of dimension expansion, our work proposes a unified framework for dimension expansion and applies it in similarity search. The framework has

---

two key components. The *optimal sparse lifting* is a sparse binary vector representation of the input samples in a higher-dimensional space, such that the pairwise similarity between the samples can be roughly preserved. The *sparse lifting operator* is a sparse binary matrix that best maps the input samples to the *optimal sparse lifting*. Computationally, both components can be efficiently and effectively obtained by solving optimization problems with the *Frank-Wolfe* method.

To verify the effectiveness of the proposed work, we carried out a series of experiments. It was found that, for given data, our approach could produce the *optimal sparse lifting* and the *sparse lifting operator* with high quality. It reported consistently and significantly improved precision and speed in similarity search applications on various datasets. And hence our work provides a solution for practical applications.

The paper is organized as follows. Section 2 reviews the related work. Section 3 introduces our model and the algorithm. Section 4 reports the empirical experiments and results, followed by the discussion and conclusion in Section 5.

## 2   Related work

### 2.1   Similarity search and locality sensitive hashing

Similarity search aims to find similar objects to a given query among potential candidate objects, according to certain pairwise similarity or distance measures [5, 21]. The complexity of accurately determining the similar objects depends heavily on both the number of candidates to evaluate and the dimension of the data [17]. Computing the similarities or distances seems straightforward, but unfortunately could often become prohibitive if the number of candidate objects is large or the dimension of the data is high.

To ensure the tractability of calculating pairwise distances for large-scale problems in high-dimensional spaces, approximate methods have to be sought, among which the *locality sensitive hashing (LSH)* method is routinely applied [11, 7, 3, 4]. The *LSH* method provides an approximate distance-preserving mapping of points from the input space to the output space. The output space usually has a much lower dimension than the input space, so that the speed of nearest neighbors search can be significantly improved.

To realize an *LSH* mapping, one common way is to compute random projections of the data samples by multiplying the input vectors with a random dense matrix of various types [3, 11]. Strong theoretical bounds exist and guarantee that the good locality can be preserved through such random projections [15, 1, 2].

### 2.2   Biological evidence of dimension expansion and the *fly* algorithm

Biological discovery in animals' neural systems keeps motivating new studies in the design of computer algorithms [16, 8, 24]. Take the fruit fly's olfactory circuit as an example. It has $d = 50$ *Olfactory Receptor Neuron (ORN)* types, each of which has different sensitivity and selectivity for different odors. The *ORNs* are connected to 50 *Projection Neurons (PNs)*. The distribution of firing rates across the *PN* types has roughly the same mean for all odors and concentrations, and therefore the dependence on the concentration disappears. The *PNs* are projected to $d' = 2,000$ *Kenyon Cells (KCs)* through sparse connections. One *KC* receives the firing rates from about six *PNs* and then sums them up [6]. With the strong feedback from a single inhibitory neuron, most *KCs* become inactive except for the highest-firing 5%. In this way a sparse tag composed of active and inactive *KCs* for each odor is generated [28, 19, 23].

The *fly* algorithm was designed by simulating the odor detection procedure of the fruit fly, which achieved quite successful results in practice [8]. Denote by $X \in \mathcal{R}^{d \times m}$ the $m$ input samples of $d$-dimensional zero-centered vectors. The inputs are mapped into hashes of $d'$ (usually $\gg d$) dimensions, by multiplying $X$ with a randomly generated sparse binary matrix $W$. Then a winner-take-all strategy is applied on the output. For each vector in $WX$, the elements with the highest $k = 100$ values are set to one, and all others zero out. In this way, a sparse binary representation (denoted by $Y \in \mathcal{R}^{d' \times m}$) in a space with a higher dimension is obtained. In short, comparing with the *LSH* method which reduces the data dimension, the *fly* algorithm increases it, while ensuring the sparsity of the data in the higher-dimensional output space.

# 3 Models

## 3.1 The *optimal sparse lifting* framework

We are interested in the problem of seeking sparse binary output vectors for given input data samples, where the output dimension is larger or much larger than the input dimension. We expect that the pairwise similarity relationship of the data in the input space can be kept as much as possible by the new vectors in the output space. Moreover, if the optimal output vectors are available for a small portion of input samples, we are also interested in the problem of approximately obtaining such a representation for other samples, but in a computationally economical way.

Mathematically, we model the two problems into a unified *optimal sparse lifting* framework as follows. Let $X \in R^{d \times m}$ be a matrix of input data samples in the $d$-dimensional space. We consider to minimize the objective function

$$f(W, Y) = \frac{1}{2} \|WX - Y\|_F^2 + \frac{\alpha}{2} \|X^T X - Y^T Y\|_F^2, \tag{1}$$

where $W \in \mathcal{R}^{d' \times d}$ and $Y \in \mathcal{R}^{d' \times m}$ are subject to some constraints; in particular, both are required to be sparse. Here, the first term aims to ensure $Y \approx WX$ [2], and the second term seeks to approximately preserve pairwise similarities between the input samples. In the function, $\alpha > 0$ is a balance parameter.

In general, we expect $d' \gg d$. Therefore, the output $Y$ is called *sparse lifting* of the input $X$, and the matrix $W$ is called *sparse lifting operator*. For simplicity of discussion, the adjective "sparse" may be dropped from time to time in the sequel.

In addition to sparsity constraint on $W$, we would like $W$ to be binary with exactly $c$ ones in each row. If we relax the binary constraint into the unit interval constraint, then $W$ should satisfy, component-wise,

$$W \mathbf{1}_d = c \mathbf{1}_{d'}, \quad 0 \le W \le 1. \tag{2}$$

Similar constraints can be imposed on $Y$ as well, for example,

$$Y^T \mathbf{1}_{d'} = k \mathbf{1}_m, \quad 0 \le Y \le 1 \tag{3}$$

with the hope that each column of $Y$ has exactly $k$ ones. But if the primary goal is to obtain a good $W$ using the training dataset, fewer constraints on $Y$ could be preferable.

Computationally, the problem formulated in Eq. (1) can be naturally solved via alternating minimization. Fix $W$ and solve for $Y$; then fix $Y$ and solve for $W$; and repeat the process. A simplified approach that performs well in practice just does one round alternating minimization using the $\ell_p \, (0 < p < 1)$ pseudo-norm to promote sparsity and binarization. Denote by $\mathcal{W}$ and $\mathcal{Y}$ the feasible regions of $W$ and $Y$ defined in Eq. (2) and Eq. (3) respectively. We solve

$$\min_{Y \in \mathcal{Y}} \frac{1}{2} \|X^T X - Y^T Y\|_F^2 + \gamma \|Y\|_p, \tag{4}$$

to get the *optimal sparse lifting* $Y_*$; then we solve

$$\min_{W \in \mathcal{W}} \frac{1}{2} \|WX - Y_*\|_F^2 + \beta \|W\|_p, \tag{5}$$

to get the *optimal lifting operator* $W_*$. Here the term "optimal" is used loosely.

We call the first step of solving Eq. (4) the *(sparse) lifting* step, and the second step of solving Eq. (5) the *(sparse) lifting operator* step.

Given the *optimal lifting operator* $W_*$, the *optimal lifting* of an input vector $x$ can be estimated by $y = (y_1, \cdots, y_{d'}) \in \{0,1\}^{d'}$ with

$$y_i = \begin{cases} 1 & if \ (W_* x)_i \ is \ among \ the \ largest \ k \ entries \ in \ W_* x; \\ 0 & otherwise. \end{cases} \tag{6}$$

**Algorithm 1** $\min_Y \left\| X^T X - Y^T Y \right\|_F^2 \quad s.t. \quad Y \in \mathcal{Y} \cap \{0,1\}^{d' \times m}$

---

1: Given $X, Y^0 \in \mathcal{Y}, \gamma^0 > 0$
2: Let $L(Y, \gamma) = \frac{1}{2} \left\| X^T X - Y^T Y \right\|_F^2 + \gamma \|Y\|_p$
3: **for** $k = 0, 1, 2, \cdots, K$ **do**
4:     Compute $S^{k+1} := \operatorname{argmin}_{S \in \mathcal{Y}} \langle S, \nabla_Y L (Y^k, \gamma^k) \rangle$
5:     Update $Y^{k+1} := \left(1 - \frac{2}{k+2}\right) Y^k + \frac{2}{k+2} S^{k+1}$
6:     Choose $\gamma^{k+1} \geq \gamma^k$
7: **end for**
8: **return** $Y^{K+1}$

---

## 3.2 Algorithm

A number of optimization algorithms are applicable to solve the two minimization problems formulated in Eq. (4) and Eq. (5). Our current study resorts to the *Frank-Wolfe* algorithm, which is an iterative first-order optimization method for constrained optimization [10, 13]. In each iteration, the algorithm considers a linear approximation of the objective function, and moves towards a minimizer of the linear function. An important advantage of the algorithm is that, for constrained optimization problems, it only needs the solution of a linear program over the feasible region in each iteration, thereby eliminating the need of projecting back to the feasible region, which can often be computationally expensive. Simple as it is, the algorithm provides quite good empirical results in our applications (ref. Section 4).

Based on the *Frank-Wolfe* algorithm, a simple iterative solution for minimizing Eq. (4) is shown in Algorithm 1. If we do not consider the increase of the balance parameter $\gamma$ in line 6, it becomes the standard *Frank-Wolfe* algorithm. In each iteration of the algorithm, the major computation comes from solving a linear program in line 4. Although in our study the linear program may involve a million or more variables, it can be solved very efficiently with modern optimization techniques [27]. The value of the balance parameter $\gamma$ increases monotonically with each iteration (e.g. $\gamma^{K+1} = 1.1 \times \gamma^K$), which makes the solution of the output matrix $Y$ tend to be sparse and binary.

From the computational complexity point of view, minimizing Eq. (5) for the *optimal lifting operator* $W_*$ is much simpler than minimizing Eq. (4). The problem can be tackled in almost the same way as in Algorithm 1. Therefore we omit the detailed discussion.

## 3.3 Optimal lifting vs. random lifting

The *fly* algorithm uses a randomly generated data transform matrix $W$ to map the dense input $X$ to $WX$ in a higher-dimensional space, followed by a sparsification and binarization process. Similarly to the *LSH* algorithm, there exists theoretical guarantee that the projection $WX$ preserves the $\ell_2$ distances of input vectors in expectation [15, 8]. However when the sparsification and binarization process is taken into consideration, no strong theoretical results are known any more.

Although motivated with the same biological characteristics in the fly olfactory circuit, our work studies the problem from a very different viewpoint. There exist two key novelties. Our work formalizes the process of the *fly* algorithm into a data-transform paradigm of *sparse lifting*. The input vectors are lifted to sparse binary vectors in a higher-dimensional space, and the feature values are replaced by their high energy concentration locations which are further encoded in the sparse binary representation.

A more significant novelty lies in the principle of projecting from the input space to the output space. The *fly* algorithm randomly generates the projection matrix $W$ and can be regarded as a *random lifting* method. Randomness exists when deciding the concentration locations due to the random generation mechanism of $W$. At the same time, although the biological connection from *Projection Neurons* to *Kenyon Cells* is still not completely clear, very recent electron microscopy images of the animal's brain have reported evidence that the connection is not random [29]. Comparatively, our proposed framework in Section 3.1 models the projection as an optimization problem which actually reduces such randomness. Along this *optimal lifting* viewpoint, many modeling and algorithmic issues could potentially arise.

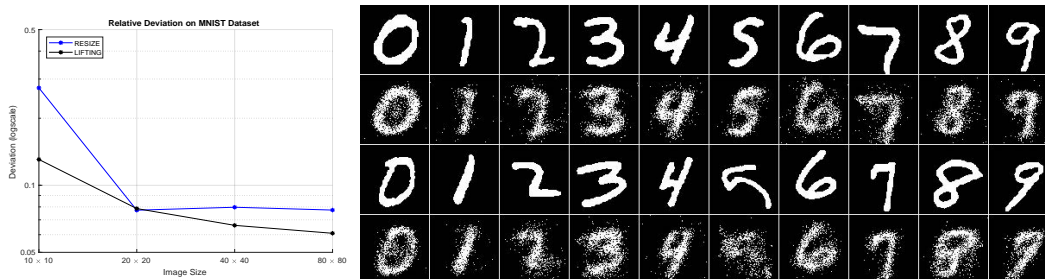

Figure 1: **The quality of the *optimal lifting* on MNIST dataset.** Left: The relative deviation from the input similarities. The *optimal lifting* (denoted by *LIFTING*) preserves the pairwise similarity even better than the *ground truth* (denoted by *RESIZE*) which were generated with the techniques of industry standard. Right: Visualization of the *lifting* results as images. The first and the third rows are the *ground truth* images with $80 \times 80$ pixels. The second and the fourth rows are the *lifting* results re-ordered by a permutation matrix.

# 4 Evaluation

## 4.1 Experimental objectives and general settings

To evaluate the effectiveness of the proposed approach, we carried out a series of experiments. Specifically, we had an experiment to illustrate the effectiveness of the *optimal sparse lifting* (ref. Section 4.2), an experiment with the same scenario of similarity search as in [8] to demonstrate the empirical superiority of the proposed *optimal lifting operator* (ref. Section 4.3), and an experiment to show the running speed comparison in a query application (ref. Section 4.4). The following benchmarked datasets were used in the experiments.

- SIFT: SIFT descriptors of images used for similarity search ($d = 128$) [14].
- GLOVE: Pre-trained word vectors based on the *GloVe* algorithm ($d = 300$) [25].
- MNIST: $28 \times 28$ images of handwritten digits in 256-grayscale pixels ($d = 784$) [18].

Besides, we also used a much larger WIKI dataset in a query application, which includes word vectors generated on the May 2017 dump of wikipedia [3] by the *GloVe* algorithm. There are $400,000$ vectors in the WIKI dataset and each vector has $500$ dimensions.

The evaluation included the empirical comparison of our work against the *fly* algorithm and the *LSH* algorithm (by random dense projection). Besides, the *autoencoder* algorithm [12] is also included in our study. An *autoencoder* is an artificial neural network used for unsupervised learning of codings. It is implemented as one hidden layer connecting one input layer and one output layer. The output layer has the same number of nodes as the input layer. An *autoencoder* is trained to reconstruct its own inputs. Usually the hidden layer has a much lower dimension than the input layer. Therefore the feature vector learned in the hidden layer can be regarded as a compressed representation of the input samples.

We implemented and tested all the algorithms on the *MATLAB* platform. Our approach used *IBM ILOG CPLEX Optimizer* as the underlying linear program solver.

## 4.2 Optimal lifting

The first experiment was carried out to evaluate the performance of the *optimal lifting* step. We hope to know if the model and the matrix factorization algorithm (ref. Algorithm 1) could well preserve the pairwise similarity between the input data samples. In the experiment, we randomly chose $5,000$ grayscale images (denoted by $X$ with each column vector $X_i$ being an image) from the MNIST dataset as the input data, and resized each image to $80 \times 80$ pixels using the cubic interpolation method, and then binarized each resized image with light pixels and dark pixels only by the Otsu's

method [22]. These $80 \times 80$ binary images generated from the industry standard techniques were regarded as the *ground truth* in this experiment, which is denoted by a matrix $G$ with each column $G_i$ being a binary image vector.

With the same set of input images, we normalized each vector $X_i$ to be of length $\sqrt{k_i}$ where $k_i$ is the number of light pixels in $G_i$. After obtaining the *optimal lifting* (denoted by $Y_*$) of these images in an $80 \times 80$-dimensional output space by Algorithm 1, we recorded the relative deviation of $\frac{\left\|X^T X - Y_*^T Y_*\right\|_F}{\|X^T X\|_F}$, and compared it with the deviation in the *ground truth* $\frac{\left\|X^T X - G^T G\right\|_F}{\|X^T X\|_F}$. Obviously, a smaller deviation value indicates a higher quality of preserving pairwise similarities between input samples.

The results are shown in Fig. 1 (left). From the results, we can see that our algorithm produced high-quality factorization results for $X^T X$. The relative deviation of the *optimal lifting* from the input is even significantly (about $20\%$) smaller than that of the *ground truth*.

Besides $80 \times 80$-dimensional images, we also tested the performance of the proposed approach with different dimensions of $10 \times 10$, $20 \times 20$ and $40 \times 40$ respectively [4]. On $40 \times 40$ images, the improvement of the relative deviation from the *optimal lifting* is very similar to that of $80 \times 80$. On $20 \times 20$ images, the *optimal lifting* is roughly similar to the *ground truth*. On $10 \times 10$ images, the improvement becomes again quite evident. The *optimal lifting* produced a relative deviation that is only half of the *ground truth*. All these results verified the effectiveness of Algorithm 1, and hence the effectiveness of the *optimal lifting* step in keeping pairwise similarities of the data.

The results of the *optimal lifting* can be visualized in an intuitive way. To do this, we computed a permutation matrix $P_*$ via minimizing $\|P Y_* - G\|_F^2$ with respect to $P$ by the *Frank-Wolfe* algorithm, and then depicted each vector in $P_* Y_*$ as a binary image. Part of the results are shown in Fig. 1 (right). In the figure, the first and the third rows are the $80 \times 80$ binary images of the *ground truth*, and the second and the fourth rows are the corresponding images from $P_* Y_*$. From the results, we can see that the *lifting* results mostly keep the shape of the images and can be recognized easily by the human being, while preserving the pairwise similarity with higher quality.

## 4.3 Similarity search

The second experiment aimed to evaluate the performance of the proposed *optimal lifting* framework in similarity search applications by comparing its accuracies against the *fly* and related algorithms. In the experiment, a subset of $10,000$ samples from each dataset were used as the testing set. All samples were normalized to have zero mean. In one run, all samples were used as a query in turn. For each query, we computed its 100 nearest neighbors among all other samples in the input space as the *ground truth*. Then we computed its 100 nearest neighbors in the output space and compared the results with the *ground truth*. The ratio of common neighbors was recorded, and averaged over all samples as the precision of each run.

For our proposed approach, we randomly selected $5,000$ different samples from each dataset as the training set. Sparse binary vectors (i.e. the *optimal lifting*) of these training samples were firstly generated with Algorithm 1 and then used to train the *optimal lifting operator* $W_*$.

For the *fly* and the *LSH* algorithms, 100 runs were carried out with randomly generated projection matrices. The mean average precision over the 100 runs and the standard deviation were recorded [20]. For the *optimal lifting* approach, only one run was executed and recorded. As a comparison, we also collected the results of the *autoencoder* algorithm (denoted by AUTOENC) [12], with which the hidden representation size is set equal to the hash length (i.e., the $k$). The *autoencoder* algorithm was trained with the same samples as our *optimal lifting* approach.

The results are depicted in Fig. 2. In all sub-figures, the horizontal axis shows different hash lengths ($k = 2, 4, 8, 16, 32$ respectively). For the *fly* and the *optimal lifting* algorithms, the output dimensions are set to $d' = 20 \times k$ and $d' = 2,000$ respectively. The vertical axis shows the one-run precisions of the *optimal lifting* and the *autoencoder* algorithms, and the mean average precisions and the standard deviations of the *fly* and the *LSH* algorithms over 100-runs. From the results it can be seen that, consistent with the results shown in [8, 26], the output vectors from the *fly* algorithm outperformed

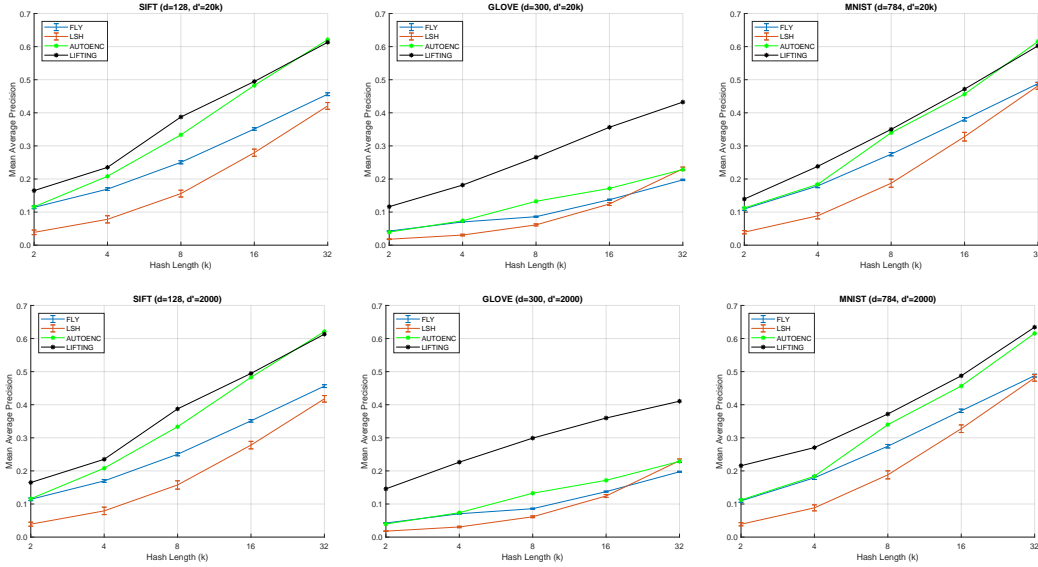

Figure 2: **Empirical comparison of similarity search precisions on different datasets.** The horizontal axis is the hash length ($k$). The vertical axis is the (mean) average precision on $10,000$ testing samples. Error bars of *fly/LSH* indicate standard deviation over $100$ trials. The embedding dimensions (not applicable to the *autoencoder* algorithm) are $d' = 20 \times k$ in the first row and $d = 2,000$ in the second row.

the vectors from the *LSH* algorithm in most experiments; while our *optimal lifting* approach reported further and significantly improved results in all experiments. The improvement on the GLOVE dataset is especially evident. All these results confirmed the benefits brought by seeking the optimal projection matrix $W_*$ instead of randomizing one.

The dense vectors generated from the *autoencoder* algorithm also improved the search precision over the vectors from the *fly* and the *LSH* algorithms on most experiments. Compare the results of the *optimal lifting* with the *autoencoder*. On SIFT and MNIST datasets, it can be seen that, when the hash length is small ($k = 2, 4, 8$), the superiority of the *optimal lifting* is quite evident. When increasing the hash length to $k = 16$ and $32$, the precisions of the *autoencoder* catch up, which tend to be quite similar as the *optimal lifting*. On GLOVE dataset, however, the improvement of our approach is still consistently significant.

## 4.4 Running speed

As a practical concern, we also measured the running time of the proposed approach, including both the training time and the query time, and compared with other algorithms. The running time was recorded on WIKI dataset with $400,000$ word vectors in $d = 500$ dimensions.

The training time of our approach includes the optimization time for both matrices $Y_*$ and $W_*$. To reduce the influence from parallel execution, only one CPU core was allowed in the experiment. The results are shown in Fig. 3 (left), and compared with the training time of the *autoencoder* algorithm. We can see that, with $5,000$ training samples and $2,000$ output dimensions, our training time is around $15$ minutes for different hash lengths ($k$), which is slower than the *autoencoder* algorithm on hash lengths of 2 and 4 but faster on hash lengths of 16 and 32. With $20 \times k$ output dimensions, our approach runs magnitude faster than the *autoencoder* algorithm on all hash lengths.

The query time was measured by searching for 100 nearest neighbors out of the $400,000$ words for $10,000$ query words with one CPU core. We reported the total query time on the output vectors of the *LSH*, *autoencoder* and *optimal lifting* algorithms respectively. As a baseline, the query time in the original input space is also shown (denoted by *NO_HASH*). From the results in Fig. 3 (right), we can see that the vectors from the *optimal lifting* approach reported significantly better speed over the

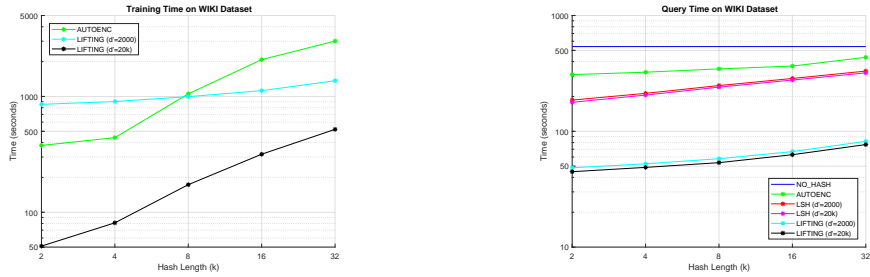

Figure 3: **Comparison of training and query time of the algorithms on WIKI dataset with** $5,000$ **training samples and** $10,000$ **query samples with a single CPU core.** Left: training time; right: query time. The horizontal axis is the hash length ($k$). The vertical axis is the time in seconds. The embedding dimension is set to $d' = 20 \times k$ and $d' = 2,000$ respectively.

others. It is magnitudes faster than searching in the original input space, and $4$ to $9$ times faster than the vectors from the *LSH* and the *autoencoder* methods.

Considering the benefits of improved query precision and speed, the cost of computing the *optimal lifting* and training the *optimal lifting operator* in our framework should be an acceptable overhead in practical applications.

## 5    Conclusion

Fundamentally different from classical approaches that seek to reduce the data dimension for analysis, our work promotes a general method for dimension expansion by a type of data transform called *optimal sparse lifting*. In this transform, feature vectors of a dataset are lifted to sparse binary vectors in a higher-dimensional space, and feature values are replaced by their "high energy concentration" locations that are encoded in the sparse binary vectors. Our proof-of-concept experiments in similarity search indicate that the proposed approach can significantly outperform, in terms of accuracy, the *random sparse lifting* and the *locality sensitive hashing* methods.

Many modeling and algorithmic issues still remain to be studied for the proposed framework, as promising as it appears to be. In addition, there are strong potentials to extend *sparse lifting* transforms to other tasks in unsupervised learning and pattern recognition, in particular to clustering analysis and data classification. To deepen understanding, further work will be necessary to study and compare the proposed approach with existing methodologies.

### Acknowledgments

This work was supported by Shenzhen Fundamental Research Fund (JCYJ20170306141038939, KQJSCX20170728162302784, ZDSYS201707251409055), Shenzhen Development and Reform Commission Fund, and Guangdong Introducing Innovative and Entrepreneurial Teams Fund (2017ZT07X152), China.

## Footnotes

[2]We may also consider to enforce $Y \approx \mu W X$ instead where $\mu > 0$ is a scaling parameter.

[3] https://dumps.wikimedia.org/

[4]For the $10 \times 10$ and $20 \times 20$ experiments, the dimension is actually reduced and it can't be called as "lifting". However, it does not prevent us from testing the algorithm's performance under these settings.

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
