[Reviews · NeurIPS 2018]

Reviewer 1



The paper studies the method of hashing a vector inspired by the "fly algorithm" where flies apparently lift an input vector in the olfactory system (after mean shift) into a higher dimensional space and then sparsify the resulting lifted vector by taking its top k coordinates. Earlier work[ 8] shows that the fly algorithm is a strong alternative to random hyperplane based LSH for similarity search. In this work they provide an algorithm to compute the optimal projection matrix instead of using a random one so that the similarity matrix preserved after the lift. They show experimentally that it outperforms autoencoder, random non optimized fly algorithm, standard LSH for similarity search. The approach is intriguing and the inspiration from fly visual cortex is very interesting. But the theme of the paper overlaps with the previous work [8] and the main novelty here is the optimization of the projection matrix. The writing and flow of the presentation could use some improvement. First to convey the value of the fly algorithm it would be useful to briefly state the exact gains from [8] of this algorithm over standard LSH early on rather than the reader having to reach the experiment section. Randomized vs optimal lifting: I am also curious if the fly's cortex is believed to use random or "trained" lifting-projection matrix W. If the intent here is to learn a better W to improve upon a random one then you should mention this point up front. In section 4.2, I found the experimental evaluation metric and the ground truth G to be a bit confusing. Why is resizing the images using cubic interpolation the right baseline? Shouldn't you instead check something like how well the output hash can predict the digit. Also, what was the value of k that was used? In comparing autoencoder, LSH and optimal lift/fly, you should also take into account the total number of bits used to represent the final hash values: Besides the number of hashes you also need to take into account the number of bits required for each hash. For your algorithm it will be log_2(d'). How much was it for LSH. I couldn't find where \mathcal{Y} was defined: is it the set of k-sparse vectors?

Reviewer 2



The paper is a natural extension to a very fundamental observation by Dasgupta, Stevens, and Navlakha in [8]. This is useful in nearest neighbour search comes from studying the olfactory circuit of a fruit fly. The observation in [8] is that using sparse binary random projections while increasing the dimensionality of the data gives nearest neighbour results that are comparable to that obtained using LSH (locality sensitive hashing) that uses dense gaussian random projections while lowering the dimension. In this paper the authors suggest computing a sparse binary projection while maintaining distances from the training data and then using this projection while performing nearest neighbour searches. They show accuracy improvements and running time comparisons using experiments with a few datasets. Quality: The overall quality of the paper is good. It discusses a natural extension for the observation made in [8]. The heuristic to compute the sparse projection is explained clearly. Clarity: The theoretical part of the paper is clearly explained though I find that section 2.2 is from previous work and might have been avoided to save space. In the experimental section, it is not clear to me what “Hash Length(k)” in the figures means. Also, when you say 20k, does it mean that it is 20 times k the hash length. Originality: The paper can be seen as an extension of the previous work [8]. The authors set up an optimisation problem satisfying the high level observations in [8]. They suggest known techniques of alternating minimisation to the solve the optimisation problem and then perform experiments. There are no new techniques, explanations, or observations (except that the heuristic works well for the analysed datasets). Significance: I think the observation in [8] provides a new direction to nearest neighbour search and I would consider the work in this paper significant as it further explores this new direction.

Reviewer 3



The paper considers the interesting question of sparse lifting and shows that in some settings, given the same computational resources, the sparse lifting method outperforms dimensionality reduction methods. The authors build on the ideas of [8] and show that in contrast to [8] which used random matrices, the "optimal" embedding based on even Frank-Wolfe procedure provides a better algorithm. The paper is well written and interesting. One comment (for which the review was not more positive than its current rating) is the issue of error correcting codes. Is it the case that paper is showing that random codes (from random lifting) are not as good as designed codes (and in fact Frank-Wolfe would suggest some form of Linear program decodable codes). This need not be the case, but that discussion is warranted. At the same time, if error correcting codes were outperforming dimensionality reduction based embeddings, then that is still a valuable fact for researchers to know (and hence the positive rating).